# In Vitro and In Silico Biological Studies of 4-Phenyl-2-quinolone (4-PQ) Derivatives as Anticancer Agents

**DOI:** 10.3390/molecules28020555

**Published:** 2023-01-05

**Authors:** Yi-Fong Chen, Bashir Lawal, Li-Jiau Huang, Sheng-Chu Kuo, Maryam Rachmawati Sumitra, Ntlotlang Mokgautsi, Hung-Yun Lin, Hsu-Shan Huang

**Affiliations:** 1Graduate Institute of Cancer Molecular Biology and Drug Discovery, College of Medical Science and Technology, Taipei Medical University, Taipei 11031, Taiwan; 2Ph.D. Program for Cancer Biology and Drug Discovery, College of Medicine, China Medical University and Academia Sinica, Taichung 40402, Taiwan; 3Ph.D. Program for Cancer Molecular Biology and Drug Discovery, College of Medical Science and Technology, Taipei Medical University and Academia Sinica, Taipei 11031, Taiwan; 4UPMC Hillman Cancer Center, University of Pittsburgh, Pittsburgh, PA 15232, USA; 5Department of Pathology, University of Pittsburgh, Pittsburgh, PA 15213, USA; 6School of Pharmacy, China Medical University, Taichung 40402, Taiwan; 7Chinese Medicinal Research and Development Center, China Medical University Hospital, Taichung 40402, Taiwan; 8Cancer Center, Wan Fang Hospital, Taipei Medical University, Taipei 11031, Taiwan; 9TMU Research Center of Cancer Translational Medicine, Taipei Medical University, Taipei 11031, Taiwan; 10Traditional Herbal Medicine Research Center of Taipei Medical University Hospital, Taipei Medical University, Taipei 11031, Taiwan; 11Pharmaceutical Research Institute, Albany College of Pharmacy and Health Sciences, Rensselaer, NY 12144, USA; 12Graduate Institute of Medical Sciences, National Defense Medical Center, Taipei 11490, Taiwan; 13Ph.D. Program in Drug Discovery and Development Industry, College of Pharmacy, Taipei Medical University, Taipei 11031, Taiwan

**Keywords:** 4-phenyl-2-quinolone (4-PQ), anticancer, tubulin, antimitotic agent, structure–activity relationship (SAR)

## Abstract

Our previous study found that 2-phenyl-4-quinolone (2-PQ) derivatives are antimitotic agents, and we adopted the drug design concept of scaffold hopping to replace the 2-aromatic ring of 2-PQs with a 4-aromatic ring, representing 4-phenyl-2-quinolones (4-PQs). The 4-PQ compounds, whose structural backbones also mimic analogs of podophyllotoxin (PPT), maybe a new class of anticancer drugs with simplified PPT structures. In addition, 4-PQs are a new generation of anticancer lead compounds as apoptosis stimulators. On the other hand, previous studies showed that 4-arylcoumarin derivatives with 5-, 6-, and 7-methoxy substitutions displayed remarkable anticancer activities. Therefore, we further synthesized a series of 5-, 6-, and 7-methoxy-substituted 4-PQ derivatives (**19**–**32**) by Knorr quinoline cyclization, and examined their anticancer effectiveness. Among these 4-PQs, compound **22** demonstrated excellent antiproliferative activities against the COLO205 cell line (50% inhibitory concentration (IC_50_) = 0.32 μM) and H460 cell line (IC_50_ = 0.89 μM). Furthermore, we utilized molecular docking studies to explain the possible anticancer mechanisms of these 4-PQs by the docking mode in the colchicine-binding pocket of the tubulin receptor. Consequently, we selected the candidate compounds **19**, **20**, **21**, **22**, **25**, **27**, and **28** to predict their absorption, distribution, metabolism, excretion, and toxicity (ADMET) profiles. Pharmacokinetics (PKs) indicated that these 4-PQs displayed good drug-likeness and bioavailability, and had no cardiotoxic side effects or carcinogenicity, but we detected risks of drug–drug interactions and AMES toxicity (mutagenic). However, structural modifications of these 4-PQs could improve their PK properties and reduce their side effects, and their promising anticancer activities attracted our attention for further studies.

## 1. Introduction

Chemotherapy is the most common remedy for localized and metastatic cancers, which uses a single drug or a combination of drugs to kill rapidly growing cancer cells [1]. Microtubules are an important chemotherapeutic target, and the anticancer activities afforded by tubulin inhibition from various small-molecular drugs were assessed [2,3]. Based on our previous studies of 4-phenyl-2-quinolone (4-PQ) analogs as microtubule inhibitors [4,5], we continued to focus on discovering novel 4-PQ derivatives in our current work.

Microtubules are composed of α- and β-tubulin heterodimers [2,3,6,7], and their main functions are to form the skeleton of cells, maintain cell morphology, fix and support the position of organelles, help the transport of intracellular substances, and assist in the movement of organelles, such as the separation of chromosomes, which are moved by microtubule traction. Anticancer drugs in nature against microtubules can be divided into two categories according to their mechanism of action [2,3,6,7]. One mechanism of antimitotic agents is to inhibit microtubule polymerization (assembly), known as destabilizing binders [2,3,6,7], such as colchicine, steganacin, podophyllin-lignan analogs (podophyllotoxin; PPT), Vinca alkaloids (vinblastine and vincristine), and combretastatins (combretastatin A-4; CA-4, and CA-2). The other antimitotic agent mechanism is to promote microtubule polymerization and inhibit microtubule depolymerization, which is referred to as stabilizing binders [2,3,6,7], such as taxanes (paclitaxel and docetaxel), laulimalide, and epothilone.

As shown inFigure 1, the methylenedioxy moiety is commonly found in various antimitotic agents, such as steganacin, cornigerine, combretastatin A-2, and PPT. In previous studies of PPT’s structure–activity relationships (SARs), the structural complexity of its four asymmetric carbon centers (chiral centers) increased the difficulty of its chemical synthesis [8]. Previous research reported that PPT is metabolized in vivo, and the *tras*-lactone on the D ring is converted to inactive *cis*-lactone by epimerization [9]; therefore, many simplified PPT analogs have been developed in recent years, such as 4-aza-PPT [10,11,12,13] and podophyllic aldehyde [14,15]. Importantly, these simplified PPT analogs still have excellent anticancer activities (IC_50_ < 1 μM) at the nanomolar scale.

Since 1993, Professor Kuo and coworkers published compounds with 2-phenyl-4-quinolone (2-PQ) skeletons as effective antimitotic agents [16,17,18]. Among synthesized 2-PQ derivatives, compound **CHM-1**, 2′-fluoro-6,7-methylenedioxy-2-phenyl-4-quinolone (Figure 1), showed good anticancer activity (log GI_50_ < −7.0) against numerous cancer cell lines by panel screening of NCI-60 human cancer cell lines [17]. **CHM-1** also demonstrated inhibition of the polymerization of microtubules [19,20]. Therefore, we continued to investigate 2-PQ analogs and applied a scaffold hopping strategy to transfer the phenyl ring at the 2-position of 2-PQ to the 4-position of the quinoline core structure, thereby obtaining a geometric isomer of 2-PQ, representing 4-phenyl-2-quinolones (4-PQs) (Figure 1) [4]. Consequently, a class of 6,7-methylenedioxy-4-substituted phenyl-2-quinolone derivatives was designed and synthesized, and their cytotoxicity against cancer cells was evaluated. These 4-PQ compounds mimic the structure of PPT analogs and serve as one of the simplified asymmetric carbon center structures of PPT [4]. Among these 4-PQ derivatives, **HPK** (2′,4′-dimethoxy-6,7-methylenedioxy-4-phenyl-2-quinolone) displayed significant antiproliferative activity in HL-60, Hep3B, and H460 cancer cells, and its IC_50_ values ranged 0.4~1.0 μM. Moreover, **HPK** treatment induced apoptosis and resulted in G_2_/M arrest through downregulating cyclin B1 and cyclin-dependent kinase 1 (CDK1) [4,21].

Because the skeleton of 4-PQs shows remarkable anticancer activities [4,21], we attempted to further develop 4-PQ derivatives. As shown in Figure 2, previous reports demonstrated that several synthetic 4-arylcoumarins (neoflavonoids, 4-phenylcoumarin compounds **A**–**D**) could inhibit the polymerization of microtubules and induce cancer cell apoptosis [22,23,24,25]. In particular, 4-arylcoumarin (**C**) was isolated from *Exostema* species (of the family Rubiaceae) and displayed cytotoxic activity against various cancer cell lines [26]. These neoflavones are abundant components that widely exist naturally in plants and have anticancer properties [27,28]. Studies reported that microtubule inhibition by 4-arylcoumarin derivatives was similar to that of combretastatin A-4, both of which bind to the colchicine-binding site [29,30]. Previous studies showed that these 4-arylcoumarin derivatives have 5-, 6-, and 7-methoxy substitutions on the coumarin core which exhibited good anticancer activity. The 2-quinolone core is the bio-isosteric structure of coumarin (Figure 2). Consequently, these results encouraged us to replace the oxygen atom (*O*) on the coumarin nucleus of 4-arylcoumarin with a nitrogen atom (*N*) by applying the concept of bio-isosterism drug design [31]. Since our previously synthesized 6,7-methylenedioxy-4-phenylquinolin-2(1*H*)-one (4-PQ) derivatives used 2-quinolone as the parent nucleus and displayed excellent anticancer activities [4], we attempted to synthesize further modified molecules of 4-PQs. According to the example of these 4-arylcoumarins having methoxy groups at positions 5, 6, and 7 with potential anticancer activities, a series of 4-PQ derivatives with 5-, 6-, and 7-methoxy groups were optimized and synthesized. Collectively, we established more-complete structure–activity relationships (SARs) of 4-PQ derivatives. In addition, we conducted in-silico studies to assess these 4-PQs by molecular modeling of αβ-tubulin inhibition and pharmacokinetic (PK) predictions.

## 2. Results and Discussion

### 2.1. Chemistry

The synthetic route of the target compound 4-PQ derivatives **19**–**32** was based on our previous methods [4] and illustrated in Figure 1. Initially, various substituted acetophenones were used as starting materials, and those acetophenones **1a**–**e** were reacted with a diethyl carbonate (**2**) to obtain corresponding benzoylacetates **3a**–**e**. Compounds **3a**–**e** were then condensed with a substituted aniline **4a**–**d** to generate corresponding benzoylacetanilide intermediates **5**–**18**. Eventually, the target compound 4-phenylquinolin-2(1*H*)-one derivatives **19**–**32** were obtained through Knorr quinoline cyclization (acid catalysis) [4,32,33]. This synthesis method utilized intermediate benzoylacetanilides **5**–**18** and heat with polyphosphoric acid (PPA) at 100–110 °C to obtain the corresponding 4-PQ compounds **19**–**32** by intramolecular cyclization. Synthetic 4-PQs were characterized by infrared (IR), and ^1^H nuclear magnetic resonance (NMR) and ^13^C NMR, and mass spectrometry, and the results of the spectrum are shown in Appendix A).

The ^1^H NMR spectra of target compounds **19**–**32** exhibited a characteristic broad peak in the range of 11.56–11.90 ppm which can be attributed to the NH of 2-quinolone and a single peak in the region of 5.86–6.39 ppm which can be attributed to H-3 of the ethylene proton in the 2-quinolone core structure. For instance, the ^1^H NMR spectrum of compound **22** was characterized by the appearance of single peaks at 3.84 and 3.86 ppm, which were attributable to the methoxy groups of the 2-quinolone moiety and 4-phenyl ring. The characteristic H-3 signal of 2-quinolone appeared at 6.38 ppm. The formation of 4-phenyl-2-quinolone was also confirmed by ^13^C NMR studies. N-C=O signals in the 2-quinolone core structure of **19**–**32** were detected within 159.94–161.92 ppm.

The detailed 2D-NMR spectral data of target compounds are provided in Appendix A. For example, the full 2D assigned correlations of compound **22** are shown in Table 1. The heteronuclear single quantum correlation (HSQC) experiment is frequently used in NMR spectroscopy to determine proton–carbon single-bond correlations. The ^1^H singlet at δ_H_ = 6.38 ppm gave an HSQC correlation with its attached carbon at δ_C_ = 106.52 ppm, assigned as H-3 and C-3. The ^1^H doublets at δ_H_ = 7.51 ppm gave an HSQC correlation with its attached carbons at δ_C_ = 104.27 ppm, assigned as H-5 and C-5 of the 2-quinolone nucleus. The heteronuclear multiple bond correlation (HMBC) experiments give correlations between carbons and protons separated by two, three, and sometimes, in conjugated systems, four bonds. The C-5 (δ_C_ = 104.27 ppm) gave an HMBC correlation with a 1H double doublet at δ_H_ = 7.32 ppm, assigned as H-7 of the 2-quinolone nucleus. The H-7 (δ_H_ = 7.32 ppm) showed an HSQC correlation at δ_C_ = 122.63 ppm, assigned as C-7. Correlated spectroscopy (COSY) is used to determine which signals arise from neighboring protons and the cross-peaks presented by coupled protons. COSY data revealed correlations between H-7 (δ_H_ = 7.32 ppm) and δ_H_ = 7.76 ppm, assigned as H-8. The ^1^H doublet at δ_H_ = 7.76 ppm gave an HSQC correlation with its attached carbon at δ_C_ = 121.18 ppm, assigned as H-8 and C-8. Long-range correlations in the HMBC experiment were observed between methoxy groups (H-9, δ_H_ = 3.84 ppm) and C-6 (δ_C_ = 156.08 ppm), and HMBC correlations of H-5 (δ_H_ = 7.51 ppm) with C-6 (δ_C_ = 156.08 ppm) and C-7 (δ_C_ = 122.63 ppm) and C-8a (δ_C_ = 156.08 ppm), of H-7 (δ_H_ = 7.32 ppm) with C-5 (δ_C_ = 104.27 ppm), C-6 (δ_C_ = 156.08 ppm), and C-8a (δ_C_ = 156.08 ppm), and of H-8 (δ_H_ = 7.76 ppm) with C-6 (δ_C_ = 156.08 ppm), C-8a (δ_C_ = 156.08 ppm), and C-4a (δ_C_ = 126.20 ppm). The 4-phenyl ring assignments were as follows: the 1H triplet at δ_H_ = 7.48 ppm gave an HSQC correlation with its attached carbon at δ_C_ = 130.57 ppm, assigned as H-5′ and C-5′. The COSY data revealed that the H-5′ triplet at δ_H_ = 7.48 ppm revealed correlations between δ_H_ = 7.36–7.39 ppm and δ_H_ = 7.12 ppm, assigned as H-4′ and H-6′, respectively. Moreover, the δ_H_ = 7.36–7.39 ppm contained another proton signal assigned as the remaining proton (H-2′). The H-6′ signal δ_H_ = 7.12 ppm has an HSQC correlation with δ_C_ = 116.31 ppm, assigned as C-6′. The 2H multiplet at δ_H_ = 7.36–7.39 ppm gave HSQC correlations of H-2′ and H-4′ with its attached carbon at δ_C_ = 113.11 and 119.92 ppm, respectively, assigned as C-2′ and C-4′, and was further confirmed by the HMBC spectrum. The HMBC experiment revealed correlations of the methoxy groups (H-10, δ_H_ = 3.86 ppm) with C-3′ (δ_C_ = 159.94 ppm), as well as HMBC correlations of H-4′ (δ_H_ = 7.36–7.39 ppm) with C-2′ (δ_C_ = 113.11 ppm) and C-6′ (δ_C_ = 116.31 ppm), of H-5′ (δ_H_ = 7.48 ppm) with C-3′ (δ_C_ = 159.94 ppm) and C-1′ (δ_C_ = 136.16 ppm), of H-6′ (δ_H_ = 7.12 ppm) with C-2′ (δ_C_ = 113.11 ppm) and C-4′ (δ_C_ = 119.92 ppm), and of H-2′ (δ_H_ = 7.36–7.39 ppm) with C-4′ (δ_C_ = 119.92 ppm), C-6′ (δ_C_ = 116.31 ppm), and C-4 (δ_C_ = 149.47 ppm).

### 2.2. Anticancer Activities and SAR Studies of New 4-PQ Derivatives

We first screened the synthesized 4-PQ **19**–**32** to identify the pharmacology of the anticancer activities against four human cancer cell lines, including the COLO205 (colorectal adenocarcinoma), A498 (renal cell carcinoma), H460 (non-small-cell lung cancer), and Hep 3B (liver cancer) cell lines. The cell viability of cancer cells was evaluated by a 3-(4,5-dimethylthiazol-2-yl)-2,5-diphenyltetrazoliun bromide (MTT) assay to determine cell proliferation inhibition. Results are presented as the 50% inhibition of cancer cell growth based on the drug concentration (IC_50_), and the preliminary results of antiproliferation are shown in Table 2.

Almost all compounds of the class, 5,6,7-methoxy-substituted 4-phenylquinolin-2(1*H*)-one (**19**–**32**), had IC_50_ anticancer activities of more than 50 μM. However, these 4-PQ derivatives showed potential anti-proliferation against the COLO205 and H460 cell lines with IC_50_ values ranging 0.32–47.1 μM. In particular, 6-methoxy-4-(3′-methoxy phenyl)quinolin-2(1*H*)-one (**22**) showed excellent anticancer activity, which was highly active against the COLO205 (IC_50_ = 0.32 μM) and H460 cell lines (IC_50_ = 0.89 μM). In addition, 4-(2′,4′-dimethoxyphenyl)-5,7-dimethoxy quinolin-2(1*H*)-one (**27**) showed moderate anticancer activity against the COLO205 cell line (IC_50_ = 7.85 μM). Interestingly, a previous study indicated that the 3,4,5-trimethoxyphenyl moiety is commonly employed in antimitotic agents [34]; however, our results showed that a tri-methoxy substitution on 4-PQs (**29**–**32**) seemed to have no anticancer effects (IC_50_ > 50 μM). Among all of the synthesized compounds, compound **22** was more effective than the parent compound, **HPK** (IC_50_ = 7.4 μM), with an IC_50_ of 0.32 μM against COLO205 cells. Therefore, the mechanism of the anticancer activity of compound **22** deserves further investigation. On the other hand, out of the 4-PQ compounds exhibiting anticancer activities (with IC_50_ values of <50 μM), compounds **19**, **20**, **21**, **22**, **25**, **27,** and **28**, were subjected to further in silico molecular docking studies and pharmacokinetic (PK) predictions.

The preliminary results of anti-proliferation indicated that the active compounds **19**, **20**, **21**, **22**, **25**, **27,** and **28** only demonstrated anticancer activities in COLO205 and H460 cells (other cell lines were shown IC_50_ > 50 μM). As shown in Table 3, we further investigated the selectivity index (SI) to analyze the sensitivities of each compound against COLO205 and H460, respectively. Compound **22** exhibited the highest sensitivity (SI = 66.04) toward COLO205 cells, while compounds **27** and **HPK** showed moderate selected ratios (SI are 2.69 and 2.86). Other compounds (**19**, **20**, **25**, and **28**) displayed no sensitivities (SI < 1) compared with **22**, **27** as well as **HPK** in COLO205. Similar results were shown in H460 cells, compound **22** had the highest selected ratio (SI = 33.64). Notably, compound **HPK** also displayed high sensitivity (SI = 33.27) against H460 cells. However, compounds **19**, **20**, **21**, **25,** and **28** showed no selectivity (SI < 1) in H460 cells. Overall, compound **22** is the most active and sensitive within these 4-PQ derivatives against both COLO205 and H460 cell lines.

### 2.3. Molecular Modeling of 4-Phenyl-2-quinolone (4-PQ) Derivatives Proposes Their Docking into the Colchicine-Binding Site

As previously reported, the cytotoxic activity of 4-phenyl-2-quinolone (4-PQ) [35,36] and its 4-arylcoumarin analog [24,25,29,30] are well-known as antimitotic agents associated with their ability to inhibit tubulin formation via binding of compounds to the colchicine-binding site. In this study, molecular docking was simulated to investigate the binding modalities of 4-PQ derivatives against tubulin using AutoDock4 [37] and AutoDock Vina software [38]. In 2004, the co-crystal structure of αβ-tubulin complexed with N-deacetyl-N-(2-mercaptoacetyl)colchicine (DAMA-colchicine, an analog of colchicine) was discovered by X-ray analysis (PDB ID: 1SA0) [39]. At the same time, the X-ray structure of the tubulin/PPT complex also revealed that PPT inhibits tubulin polymerization by blocking the colchicine-binding site (PDB ID: 1SA1) [39] and presents a similar orientation as that of DAMA-colchicine. Accordingly, as shown in Figure 3A, molecular docking experiments were performed on the 4-PQ series and their virtual binding interactions with α- and β-tubulin (PDB ID: 1SA1), and results were compared to PPT and colchicine as reference drugs. To validate docking protocols, the co-crystallized ligand PPT and colchicine were re-docked into the colchicine-binding site of isolated α- and β-tubulin proteins. As shown in Figure 3B,C, co-crystallized poses of the reference drugs were superimposed on the re-docking pose. PPT and colchicine could be re-docked to their respective target proteins with root mean squared deviation lower bounds (RMSD/lbs) of 0.017 (Figure 3B) and 1.879 Å (Figure 3C) via AutoDock Vina. Additionally, the re-docked conformations of PPT and colchicine had similar orientations to their respective co-crystallized ligands. The docked poses of the 4-PQs are superimposed with PPT and colchicines in Figure 3D. Results showed that the methoxy groups on the 4-phenyl ring (C ring) of 4-PQ were located in the tri-methoxy groups’ position of PPT and colchicines (as indicated by yellow arrows in Figure 3D). Additionally, the methoxy groups on the 2-quinolone core (A ring) of 4-PQ were located in the methylenedioxy moiety of PPT and the ketone group of colchicines (as indicated by red arrows).

We selected some potential 4-PQ compounds, representing **19**, **20**, **21**, **22**, **25**, **27,** and **28**, which had IC_50_ values of <50 μM by the MTT assay to predict their binding modalities into the colchicine site. Results suggested that these 4-PQ derivatives could snugly occupy the colchicine-binding site of αβ-tubulin (Figure 3A). On the other hand, a previous report on deciding which program to use in docking studies indicated that Vina is much faster and adopts more accurate binding poses but Autodock4 forms better binding affinity [40]. Therefore, we executed both docking programs to estimate the binding energies, and the possible docking poses were predicted via the AutoDock Vina. The binding affinity of each 4-PQ attached to αβ-tubulin is discussed based on the estimated ΔG value, and results are given in Table 4. The binding energies of these compounds were in the range of −7.46 to −9.91 (kcal/mol) via AutoDock4 calculation, while the range of Vina estimation is −5.2 to −8.1 (kcal/mol). The estimated ΔG of PPT was lower (−9.91 kcal/mol in Autodock4; −9.0 kcal/mol in Vina) than the other 4-PQs and the **HPK** parent compound (−7.90 kcal/mol in Autodock4; −8.3 kcal/mol in Vina), which means that the inhibitory potency of these 4-PQs was less than that of the control drug. However, compound **22** (−7.97 kcal/mol in Autodock4; −8.1 kcal/mol in Vina) had a very close ΔG value compared to **HPK** (−7.90 kcal/mol in Autodock4; −8.3 kcal/mol in Vina).

All docked molecules interacted with amino acid residues of αβ-tubulin within the colchicine-binding active site (Figure 4, Figure 5 and Figure 6); furthermore, the results were analyzed by Discovery Studio Visualizer software and summarized in Table 5. As shown in Figure 4A, the proposed binding mode of PPT formed five carbon–hydrogen bonds with ALA250, VAL315, ALA317, LYS352, and ASN350 residues. The methylenedioxy pharmacophore formed two hydrogen bonds with VAL315 and ASN350. The carbonyl group of the lactone ring formed one hydrogen bond with ALA250, and the methoxy group of PPT’s phenyl ring (C-ring) formed two hydrogen bonds with ALA317 and LYS352. The phenyl ring (ring A) showed one pi-sulfur bond with MET259. In addition, the A-ring, C-ring, methylenedioxy moiety, and tri-methoxy groups of PPT formed several hydrophobic interactions with VAL181, CYS241, LEU242, ALA250, LEU255, ALA316, VAL318, LYS352, ALA354, and ILE378. Similar results were also seen for colchicine (Figure 4B); tri-methoxy groups, the A-ring, and C-ring of colchicines formed hydrophobic interactions with CYS241, LEU242, ALA250, LEU255, ALA316, VAL318, LYS352, ALA354, and ILE378, while the carbonyl group and methoxy group of the A-ring moiety formed four carbon-hydrogen bonds with ALA180, ASN258, VAL315, and ASN350. In addition, one methoxy group of the C-ring of colchicine showed one carbon–hydrogen bond with VAL238. Taken together, these results revealed that the methoxy groups or some hydrogen bond acceptors (containing oxygen, such as carbonyl) on the A- and C-rings of the ligand played pivotal roles with the αβ-tubulin receptor in the colchicine-binding site. Interactions with the A-ring site were delineated by the following amino acid residues: ALA180, VAL181, ASN258, MET259, VAL315, ALA316, ASN350, and LYS352. On the other hand, the ligand’s moiety in the C-ring site interacted with amino acid residues of αβ-tubulin, including residues CYS241, LEU242, VAL238, ALA250, LEU255, ALA316, ALA317, VAL318, LYS352, ALA354, and ILE378.

In the current study, we identified 4-PQ compounds with extensive interactions with amino acids of the colchicine-binding site of αβ-tubulin. The predicted binding poses of compounds **19**, **20**, **21**, **22**, **25**, **27**, **28**, and **HPK** showed respective affinity values of −7.5, −7.4, −8.1, −8.1, −6.1, −5.9, −5.2, and −8.3 kcal/mol via Vina estimation. The amide groups (carbonyl of 2-quinolone core) of compounds **19**, **21,** and **22** formed an important conventional hydrogen bond with ALA250 (Figure 5), which is similar to the carbonyl group in the γ-lactone ring (D-ring) of PPT displayed carbon hydrogen bond with β-tubulin. The A- and B-rings of the 2-quinolone moiety displayed several hydrophobic interactions with LEU248, ALA316, and LYS352 (Figure 6A–D). In addition, the amide groups (NH) of compound **28** formed a conventional hydrogen bond with THR179 (Figure 6C). These findings implied that the 2-quinolone core structure plays an important role in the colchicine-binding domain, which is consistent with our previous study of 4-benzyloxy-2-quinolone derivatives on tubulin inhibition [5]. Each of their A- and C-rings revealed similar hydrophobic interactions with αβ-tubulin, which were just like those of PPT and colchicine.

### 2.4. In Silico Prediction of Drug-Likeness Studies

Drug-likeness is considered an important tool for predicting whether a molecule can be a potential drug candidate. Therefore, we employed SwissADME [41] to forecast the bioavailability and drug-likeness properties, and the results are shown in Table 6. Several rules were used to assess the drug-likeness, the most famous of these descriptions is Lipinski’s rule of five [42]. According to Lipinski’s rule, a compound is labeled as drug-like when it meets the following criteria: molecular weight < 500 Da, H-bond donors (HBDs; the sum of OHs and NHs) ≤ 5, H-bond acceptors (HBAs; the sum of Ns and Os) ≤ 10, and Log P ≤ 5. The Log P values are measured as the partition coefficient between n-octanol and water and represented the molecular hydrophobicity or lipophilicity. High Log P values indicate poor absorption or low permeability, while low Log P values indicate increased absorption and permeability. Log P values of all predicted compounds were <5 (ranging from 2.25–3.05), which is within an acceptable range. The molecular polar surface area (PSA) is a very useful parameter for predicting drug transport properties. The PSA is defined as the sum of surface polar fragments (usually oxygens, nitrogens, and attached hydrogens) in a molecule. The topological PSA (TPSA) provides results of practically the same quality as the classical three-dimensional PSA; the calculations, however, are two to three orders of magnitude faster [43]. The 4-PQ compounds had TPSA values ranging from 32.86–69.78 Å^2^, which is a good range for oral bioavailability since most therapeutic molecules have TPSA values of <140 Å^2^ for good passive membrane permeability (in a non-polar environment) [44].

Moreover, none of the 4-PQs were predicted as pan assay interference compounds (PAINS) [45], which suggests the absence of potential pan-assay interferences or being frequent hitters of promiscuous compounds. On the other hand, we found no violations of Lipinski’s [42], Veber’s [44], Ghose’s [46], Egan’s [47], or Mugge’s [48] rules for any of the synthesized 4-PQ compounds based on the obtained data.

### 2.5. In Silico Analysis of Potential PK (ADMET) and Toxicological Properties (T)

It is clear that in addition to pharmacological properties, ADMET (absorption, distribution, metabolism, elimination, and toxicity) research plays a crucial role in the success of drug candidates [49,50]. Because of the impact on eventual success, such studies now occur early in the drug discovery process [51]. The PK characteristics of 4-PQ compounds (**19**, **20**, **21**, **22**, **25**, **27,** and **28**), PPT, and colchicine were assessed using online SwissADME software [41] and admetSAR [52]. As shown in Table 7, all compounds were predicted to have high gastrointestinal (GI) absorption (human intestinal absorption; HIA). In addition, all of the 4-PQ derivatives were predicted to have the ability to pass through the blood–brain barrier (BBB), except for PPT and colchicine. P-glycoprotein (P-gp), a member of ATP-binding cassette (ABC) transporters, is an efflux transporter that can pump drugs out of cells to reduce the biological effects of anticancer drugs [53], which may cause a failed treatment. If a drug under investigation is identified as a P-gp substrate, this means that its concentration levels in cells may be clinically affected by a P-gp inhibitor. On the other hand, P-gp plays a key role in multidrug resistance (MDR) in cancer [54]. Most of these were not P-gp substrates, as only **HPK** and colchicine were predicted to have potential interference with P-gp.

Cytochrome P450 (CYP450) enzymes are pivotal mediators in the metabolism of many medicines. The CYP450 expression level determines a drug’s metabolic rate. Unanticipated drug–drug interactions (DDIs) and drug metabolism problems based on CYP450 enzymes are also common causes of adverse drug events (ADEs) [55]. In the metabolic profiles, all of these compounds served as different kinds of CYP450 enzyme inhibitors (Table 7), suggesting that they may compete with other drugs, which means that they can block other drugs’ metabolic activity to extend the half-life of therapeutic drugs in patients’ serum levels. Further PKs were investigated in the future for any potential drug–drug interactions. In particular, compounds **22**, **25**, **27**, **28,** and **HPK** inhibited more than three types of CYP450 enzymes, including CYP1A2, CYP2C9, CYP2D6, and CYP3A4, which reveals that they would likely interfere with the metabolism of other drugs using the same CYP450 pathway. These findings implied that using these 4-PQs should be done while being aware of the risks of CYP450 enzyme-based unanticipated DDIs and metabolism problems. On the other hand, several computational approaches can help predict the potential toxicity of new compounds in the initial stages of drug discovery, primarily considering human ether-à-go-go-related gene (hERG) inhibition, AMES toxicity, and carcinogenicity. hERG potassium channels are essential for regulating electrical activity in the human heart [56]. Inhibition of hERG potassium channels may lead to long QT syndrome (LQTS), which is known as a fatal ventricular tachyarrhythmia called torsades de pointes in clinical studies [57,58]. As shown in Table 8, the toxicity profile predicted by hERG indicated that 4-PQs were non-hERG inhibitors. Moreover, the carcinogenicity assessment showed that 4-PQ compounds were non-carcinogens. However, AMES toxicity showed that most of the 4-PQ compounds were found to be AMES toxic (mutagens), including compounds **20**, **21**, **22**, **25,** and **HPK**. These results revealed that when using these 4-PQs, one should be aware of potential DDI side effects and risks of mutagenicity. The U.S. Environmental Protection Agency (EPA) established four toxicity categories for acute hazards: class I is high toxicity and severely irritating, class II is moderate toxicity, class III is slightly toxic and low toxicity, and class IV is non-toxic [59]. All of the predicted compounds were found to have 50% lethal dose (LD_50_) values that ranged from 2.1994–3.0013 mol/kg and were classified as class III. These results implied that these 4-PQs have low acute toxicity.

The SwissADME software briefly describes six essential properties of oral bioavailability to create a bioavailability radar and boiled egg plots for analysis. These properties and suitable values are lipophilicity (LIPO): Log P (XLOGP3) value from −0.7 to 5.0; SIZE, molecular weight in the 150–500 g/mol range; polarity (POLAR), TPSA values from 20 to 130 Å^2^; insolubility (INSOLU), solubility with Log S (ESOL) from −6 to 0; insaturation (INSATU), saturation fraction of carbons with sp3 hybridization (fraction Csp3) from 0.25 to 1; and flexibility (FLEX), the number of the rotatable bonds from 0 to 9 [41]. Results are shown in Figure 7A−J with bioavailability radar for 4-PQ compounds, and the pink area in the radar is a range of optimal values. Bioavailability radars for all 4-PQ molecules exhibited suitable parameters in terms of lipophilicity, size, polarity, insolubility, and flexibility, but were not good at insaturation (INSATU). The purpose of the boiled egg plot is to correctly predict HIA and BBB properties of substances that are important for therapeutic research [60]. The blue dots represent P-gp substrates (PGP^+^) and the red dots indicate P-gp non-substrates (PGP^−^). The boiled egg forecast of 4PQs is shown in Figure 7K, the outer gray area is for compounds with lower GI absorption and limited BBB penetration. Furthermore, the white area represents the physicochemical space of the molecule with the highest probability of passive HIA, while the yellow area represents the physicochemical space of the molecule with the highest probability of penetrating the BBB. All of the 4-PQs exhibited predicted high BBB penetration, as demonstrated by their location inside the yellow ellipse (yolk) defining BBB absorption (Figure 7K). Essentially, all of the 4-PQs that were predicted to enter the brain also displayed a predicted high HIA (Table 7). Only colchicine and PPT were predicted to display low brain permeation but high intestinal absorption.

We further employed the site of metabolism (SOM) prediction (SOMP) [61], a web service for in silico prediction of the SOM, to gain insights into the biotransformed positions of the most effective compound **22** (the original data can be found in Appendix A). The predicted results for compound **22** are illustrated in Figure 8A. In addition, possible metabolites of compound **22** were presented by BioTransformer 3.0 [62], a web server that accurately predicts metabolic transformation products. As shown in Figure 8B, several metabolites of 4-PQ compound **22** were formed by aromatic hydroxylation on its aromatic A- and C-rings via phase I metabolism of CYP450. Typical metabolism of the methoxy group, *O*-dealkylation, was also found on compound **22**’s 6-methoxy of the 2-quinolone core and 3′-methoxy of the 4-phenyl ring. Of note, the epoxidation of an alkene was catalyzed by CYP450 to form an epoxide on ring B of compound **22**. Further phase II metabolism takes place to form OH-glucuronidation and glutathione (GSH)-conjugation of epoxide by UDP-glucuronosyltransferase and glutathione transferase, respectively. These hydrophilic conjugates and their metabolites can be excreted in phase III of their metabolism.

## 3. Materials and Methods

### 3.1. Materials and Physical Measurements

Human hepatoma (Hep 3B; HB-8064-ATCC), lung cancer cells (H460; HTB-177-ATCC), and normal skin cells (Detroit 551; CCL-110-ATCC) were purchased from American Type Culture Collection (Manassas, VA, USA) (https://www.atcc.org/; accessed on 3 December 2022). Human colon carcinoma (COLO205; CCL-222-ATCC) and renal cancer cells (A498; HTB-44-ATCC) were commissioned by the Food Industry Research and Development Institute (Hsinchu, Taiwan) to purchase. All solvents and reagents were purchased commercially and applied without additional purification. The progress of all reactions was monitored by thin layer chromatography (TLC) on 2 × 6-cm pre-coated silica gel 60 F_254_ plates of 0.25 mm thick (Merck, Darmstadt, Germany). Chromatograms were visualized under UV light at 254–366 nm. Silica gel 60 adsorbent (Merck, particle size 0.063–0.200 mm) was used for column chromatography. Melting points (MPs) were determined with a Yanaco MP-500D melting point apparatus and are uncorrected. Infrared (IR) spectra were recorded on a Shimadzu IR-Prestige-21 spectrophotometer (Kyoto, Japan) as KBr pellets. Nuclear magnetic resonance (NMR) spectra were obtained on a Bruker Avance DPX-200 (200 MHz) and DPX-400 (400 MHz) (Billerica, MA, USA) FT-NMR spectrometer at room temperature, and chemical shifts are reported in ppm (δ). The following abbreviations were used: s, singlet; d, doublet; t, triplet; dd, double doublet; and m, multiplet. The MPs, IR, and NMR were tested by the instruments of China Medical University Precision Instrument Center, Taichung, Taiwan. The mass was examined by Finnigan MAT 95S (Instrumentation Center, National Taiwan University, Taipei, Taiwan) to obtain spectra of high-resolution electrospray ionization (HRESI).

### 3.2. Chemistry

#### 3.2.1. General Procedure for the Synthesis of Benzoylacetates (**3a**–**e**)

Compound ethyl 3-oxo-3-phenyl propanoate (**3a**) was purchased commercially and used without further purification. Compounds **3b**–**e** were prepared according to our previously reported procedures [4]. A mixture of diethyl carbonate (**2**; 1 equivalent) and appropriate acetophenone (**1b**–**e**; 1 equivalent) was dissolved into 50–100 mL toluene, and then sodium hydride (NaH) (2 equivalents, 60% dispersed in mineral oil) was added. The mixture was stirred and heated to reflux for 30 min; the reaction was monitored by thin layer chromatography (TLC; with n-hexane/ethyl acetate = 10/1 as the eluent solution). After the reaction was complete, the mixture was poured into 200 mL of ice water and acidified with glacial acetic acid to pH 4–5. The mixture was extracted with ethyl acetate, dried, filtered with anhydrous magnesium sulfate (MgSO_4_), concentrated under reduced pressure, and separated and purified by column chromatography (n-hexane/ethyl acetate = 10/1) to obtain the corresponding benzoylacetates (**3b**–**e**). All synthetic compounds were in agreement with ^1^H NMR and ^13^C NMR spectroscopic data.

#### 3.2.2. General Procedure for the Synthesis of Benzoylacetanilides (**5**–**18**)

A mixture of the benzoylacetates (**3a-e**, 1 equivalent) and substituted aniline (**4a**–**d**, 1 equivalent) was stirred and heated to reflux in 50–100 mL toluene overnight. Reactions were monitored by TLC. After the reaction was complete, the mixture was cooled down to room temperature, and if a solid had formed, it was filtered by suction. If no solid had formed, then it was extracted with 10% NaOH, the aqueous layer with glacial acetic acid was acidified to pH 4–5, and the solid was separated out. The resulting precipitate was isolated by suction filtration. If no solid was found after acidification, the residue was extracted with ethyl acetate, and the organic layer was dried with anhydrous magnesium sulfate and filtered. The filtrate was concentrated under reduced pressure and purified by column chromatography to obtain the corresponding benzoylacetanilides (**5**–**18**).

#### 3.2.3. General Procedure for the Synthesis of 5-, 6-, 7-Methoxy-substituted 4-Phenyl-2-quinolones (**19**–**32**)

A 5–6 weight of polyphosphoric acid (PPA) was added to benzoylacetanilide (**5**–**18**), and the mixture was heated at 100–110 °C for 1~2 h. The mixture was cooled after TLC monitoring (*n*-hexane/ethyl acetate = 1/1 or ethyl acetate) indicated that the reaction was complete. The reaction solution was cooled to room temperature and poured into 200 mL of ice water to terminate the reaction. If a solid had precipitated, the residues were filtered with suction and recrystallized by ethanol or ethyl acetate; if no solid had precipitated, the residues were extracted with ethyl acetate, and the organic layer was dried with anhydrous MgSO_4_. The filtrate was concentrated under reduced pressure and then separated and purified by column chromatography (silica gel) to obtain the corresponding pure 4-phenylquinolin-2(1*H*)-one (**19**–**32**).

### 3.3. MTT Assay for Antiproliferative Activity

Human tumor cell lines of a cancer screening panel were maintained in RPMI-1640 medium supplemented with 10% fetal bovine serum (FBS; GIBCO/BRL), penicillin (100 U/mL)/streptomycin (100 g/mL) (GIBCO/BRL), and 1% L-glutamine (GIBCO/BRL) at 37 °C in a humidified atmosphere containing 5% CO_2_. Human hepatoma Hep 3B and normal skin Detroit 551 cells were maintained in Dulbecco’s modified Eagle medium (DMEM) supplemented with 10% FBS (GIBCO/BRL), penicillin (100 U/mL)/streptomycin (100 g/mL) (GIBCO/BRL), and 1% L-glutamine (GIBCO/BRL) at 37 °C in a humidified atmosphere containing 5% CO_2_. Logarithmically growing cancer cells were used for all experiments. Human tumor cell lines were treated with vehicle or test compounds for 48 h. According to a previous report [5], the cell growth rate was determined by an MTT (3-(4,5-dimethylthiazol-2-yl)-2,5-diphenyltetrazoliun bromide) reduction assay.

The tested compounds were initially prepared in the stock solution with a concentration of 100 μM and diluted to evaluate until found the IC_50_ values. The IC_50_ values of more than 50 μM are regarded as having no anti-proliferation effect. After 48 h of treatment, the cell growth rate was measured on an enzyme-linked immunosorbent assay (ELISA) reader at a wavelength of 570 nm, and IC_50_ values of the test compounds were calculated.

### 3.4. Molecular Docking

The crystal structure of αβ-tubulin with the co-crystallized inhibitors, N-deacetyl-N-(2-mercaptoacetyl)colchicine (DAMA-colchicine) and podophyllotoxin (PPT), was obtained from the Protein Data Bank under respective PDB codes of 1SA0 and 1SA1 [39]. The chemical structures of 4-phenyl-2-quinolone (4-PQ) derivatives were drawn, and the energy of the 3D structure was minimized using the MM2 force field [63] in ChemBio3D ultra 12.0. Molecules were docked into the colchicine-binding pocket using AutoDock4 [37] and AutoDock Vina [38]. The binding abilities were calculated as the binding free energy (ΔG, kcal/mol) in both AutoDock4 and Vina. The grid map with autogrid (for grid calculation) within MGLTools (version 1.5.6) was carried out to define interactions of the protein and ligand in the colchicine-binding site. The grid box size was built of 40 × 40 × 40 points in the x, y and z directions, and the centers of x, y and z dimensions are 117, 89, and 7.5 in AutoDock4. The Vian’s parameters were set as follows: a grid box size was 20 × 20 × 20, and the grid center was located at x = 117, y = 89, and z = 7.5. The molecular 3D-docking results were prepared and visualized using PyMOL [64] (https://pymol.org/2/; accessed on 6 May 2022). 2D-interaction maps of docked complexes were visualized with the BIOVIA Discovery Studio Visualizer (http://accelrys.com; accessed on 6 May 2022) to show interactions of compounds with amino acid residues of the target protein.

### 3.5. In Silico Predictions of Physicochemical Properties

Absorption, distribution, metabolism, excretion/elimination, and toxicity (ADMET) of drug-like PKs were calculated utilizing SwissADME [41] and admetSAR [52]. The drug-like potentials of the chosen ligands (based on the potential for in vitro antiproliferation) were estimated by observing their PK and pharmacodynamics features. SwissADME provided the parameters, including physicochemical properties (molecular weight, fraction Csp3, numbers of rotatable bonds, numbers of hydrogen bond acceptors, and numbers of hydrogen bond donors), lipophilicity (Log Po/w for iLOGP, XLOGP3, WLOGP, MLOGP, and SILICOS-IT), water solubility (Log S for ESOL, Ali, and SILICOSIT), PKs (gastrointestinal (GI) absorption, blood–brain barrier (BBB) permeant, P-glycoprotein (P-gp) substrate, CYP1A2 inhibitor, CYP2C19 inhibitor, CYP2C9 inhibitor, CYP2D6 inhibitor, and CYP3A4 inhibitor), drug-likeness rules (Lipinski, Veber, Ghose, Egan, and Muegge), and pan assay interference compounds (PAINS) methods. The admetSAR online server provided predictions of toxicity properties such as human hERG inhibition, AMES toxicity, and carcinogenicity. The sites of metabolism (SOMs) of compound **22** were predicted by the SOM prediction (SOMP) via submitting SMILES codes [61]. Possible structures of metabolites were accessed using BioTransformer 3.0 [62].

## 4. Conclusions

Previously, 2-quinolone derivatives were identified as anti-tubulin agents [5,36]. To discover novel 2-quinolone-bearing molecules of 4-phenyl-2-quinolone (4-PQ) derivatives with anticancer activities, we attempted to design and synthesize a series of methoxy-substituted 4-PQ compounds. In the current study, we investigated the structure–activity relationships of the methoxy groups at positions 5, 6, and 7 on the 2-quinolone core, and evaluated the anticancer effects of the methoxy substitutions on their 4-phenyl ring. Preliminary antiproliferative effects of compounds **19**–**32** indicated that most of the 4-PQ derivatives had IC_50_ values of >50 μM. However, some of the 4-PQs, such as compounds **19**, **20**, **21**, **22**, **25**, **27**, and **28**, showed potential anticancer effects against COLO205 and H460 cells (with IC_50_ values of <50 μM). In particular, the compound 6-methoxy-4-(3-methoxyphenyl)quinolin-2(1*H*)-one (**22**) exhibited excellent anticancer activities against COLO205 and H460 cancer cells with respective IC_50_ values of 0.32 and 0.89 μM. Computer modeling indicated that the colchicine-binding site of αβ-tubulin could deeply dock to these 4-PQs, which displayed similar orientations as those with podophyllotoxin and colchicine. The A- and C- rings of 4-PQ overlapped with those of colchicine and podophyllotoxin. In the drug discovery process, weak bioavailability and pharmacokinetics always lead to drug development failure. Of note, these 4-PQs showed suitable drug-likeness, adequate bioavailability, and non-cardiotoxicity; however, toxicity predictions indicated the risk of drug–drug interactions and carcinogenic side effects, so these 4-PQs must be used with caution. Overall, LD_50_ values of these 4-PQs were classified in class III, indicating that these 4-PQs have low acute toxicity. Therefore, further structure optimization of 4-PQs is worth investigating to convert them into clinically useful anticancer agents in the future. Altogether, our findings suggest that 4-PQ derivatives with methoxy substituents may provide new insights into these compounds in antimitotic drug development, particularly through αβ-tubulin inhibition.

## Data Availability

Not applicable.

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
