# Peer review of "In Vitro and In Silico Biological Studies of 4-Phenyl-2-quinolone (4-PQ) Derivatives as Anticancer Agents"

_molecules, 2023, doi:10.3390/molecules28020555_

Round 1

Reviewer 1 Report

The papers conserns the synthesis and computational investigation of the antitumoral activity of 4-PQ derivatives targeting tubolin as receptor.

Overall the work is well written and scienfic sounding, thare are only minor revision required.

All the docking figures 3-6 and figure 7 must be improved in quality and readibility. 

Some concerns about the docking protocol and software: why didi you use only Autodock Vina and not refine the binding energy with Autodock? Pelase add the new binding energies in the table (Autodock use a more accurate score function with respect Vina used only for HTVS of a huge number of compounds).

 Page 18: correct Autodock with autogrid (for grid calcualtion) within MGLTools ...add version...

Author Response

Dear Editor,

  On behalf of the corresponding author, Professor Lin and Huang, I am sending this mail to re-submit the revised article entitled “In Vitro and In Silico Biological Studies of 4-Phenyl-2-Quinolone (4-PQ) Derivatives as Anticancer Agents” to your Journal for publication consideration. 

  We wish to thank the reviewers for their constructive comments. Based on the reviewers' comments, we have revised the entire manuscript. The changes in the text are labeled in the tracking function of Microsoft Word. Responses to reviewers’ comments were explained in the following section below. We did clarify point-by-point in the following responses.

  We confirm that neither the manuscript nor any parts of its content are currently under consideration or published in another journal. All authors have approved the manuscript and agree with its submission to Molecules. Thank you for considering our revised manuscript.

According to the editor's and reviewers' comments, we made the amendments as follows:

Response 1: We updated the information on where the cell lines were obtained in the Materials and Methods section.

Responses to reviewer 1 comments are listed as follows.

Comments and Suggestions for Authors

The papers conserns the synthesis and computational investigation of the antitumoral activity of 4-PQ derivatives targeting tubolin as receptor.

Overall the work is well written and scienfic sounding, thare are only minor revision required.

  • All the docking figures 3-6 and figure 7 must be improved in quality and readability.

Response 2: Thanks for the reviewer's comment. We have improved the image qualities and saved the docking figures 3-7 in JPG format with high-resolution pixels.

  • Some concerns about the docking protocol and software: why didi you use only Autodock Vina and not refine the binding energy with Autodock? Pelase add the new binding energies in the table (Autodock use a more accurate score function with respect Vina used only for HTVS of a huge number of compounds).

Response 3: According to the reviewer's suggestion, we added the free-binding energies via the AutoDock4 program in the revised Table 4. A report indicated that Vina adopts more accurate binding poses but Autodock4 forms better binding affinity [Journal of Chemical Information and Modeling 2020, 60, 204-211, doi:10.1021/acs.jcim.9b00778.]. Hence, we executed both docking programs (AutoDock4 and AutoDock Vina) to estimate the binding energies, and the possible docking poses were predicted via the Vina. In addition, we added the discussion regarding the free-binding energies of AutoDock4 and Vina.

  • Page 18: correct Autodock with autogrid (for grid calcualtion) within MGLTools ...add version...

Response 4: Thanks for the reviewer's comment. We have corrected the mentions about AutoDock4 and AutoDock Vina and added the version of MGLTools (version 1.5.6) in the method of molecular docking section on page 19.

Sincerely,

First author

Yi-Fong Chen, Ph.D.

Full Current Mailing Address:

Graduate Institute of Cancer Biology and Drug Discovery

College of Medical Science and Technology

Taipei Medical University

E-mail: slugcousin7682@gmail.com

      slugcousin7682@tmu.edu.tw

Reviewer 2 Report

In my opinion, the manuscript is generally well structured and I think it contains enough original and interesting material to enrich the research so far. It requires some modifications before being published which I have outlined in the comments below:

Line 79: “Kuo et al. and coworkers……………[15-17]” - this is not a proper citation of references [15-17]

The cytotoxicity of compounds on cancerous and non-cancer cells should be compared with that of the reference anticancer drug on the same cells

What does "–" mean in Table 2 (compound 28)?

Selectivity indices (SI) for anticancer active compounds should be calculated in each cell line.

Fig. 7: The bioavailability radar graphs as well as the boiled egg prediction model are not of good enough quality to read.

Section 3.3.: The concentration range in which the compounds were tested should be given

References should be adapted to the journal's requirements.

Text formatting should be carefully checked.

The language should be modified carefully.

Author Response

Dear Editor,

  On behalf of the corresponding author, Professor Lin, and Huang, I am sending this mail to re-submit the revised article entitled “In Vitro and In Silico Biological Studies of 4-Phenyl-2-Quinolone (4-PQ) Derivatives as Anticancer Agents” to your Journal for publication consideration. 

  We wish to thank the reviewers for their constructive comments. Based on the reviewers' comments, we have revised the entire manuscript. The changes in the text are labeled in the tracking function of Microsoft Word. Responses to reviewers’ comments were explained in the following section below. We did clarify point-by-point in the following responses.

  We confirm that neither the manuscript nor any parts of its content are currently under consideration or published in another journal. All authors have approved the manuscript and agree with its submission to Molecules. Thank you for considering our revised manuscript.

According to the editor's and reviewers' comments, we made the amendments as follows:

Response 1: We updated the information on where the cell lines were obtained in the Materials and Methods section.

Responses to reviewer 2 comments are listed as follows.

Comments and Suggestions for Authors

In my opinion, the manuscript is generally well structured and I think it contains enough original and interesting material to enrich the research so far. It requires some modifications before being published which I have outlined in the comments below:

  • Line 79: “Kuo et al. and coworkers……………[15-17]” - this is not a proper citation of references[15-17]

Response 5: Thanks for the reviewer's comment. We have corrected the sentence as "Since 1993, professor Kuo and coworkers published ...... [15-17]." on page 2.

  • The cytotoxicity of compounds on cancerous and non-cancer cells should be compared with that of the reference anticancer drug on the same cells.

Response 6: Thanks for the reviewer's constructive suggestions. In Table 2, We added the GI50 (growth inhibition of 50 %; the concentration unit is molar (M)) of reference drugs (podophyllotoxin and colchicine) by the Developmental Therapeutics Program (DTP) of the National Cancer Institute (NCI) 60 cell lines screen 5 dose-response data (https://dtp.cancer.gov/). Additionally, the 5 dose-response data of podophyllotoxin and colchicine are listed in the revised Supplementary Materials. The Log (GI50) value of podophyllotoxin is -8 (0.01 μM), -8 (0.01 μM), and -8 (0.01 μM) in COLO205, A498, and H460 cells. The Log (GI50) value of colchicine is -7.6 (0.025 μM), -7.5 (0.03 μM), and -7.7 (0.02 μM) in COLO205, A498, and H460 cells.

  • What does "–" mean in Table 2 (compound 28)?

Response 7: The "–" means that the compound did not test in this cell line and the data were not determined in Table 2. In addition, we have corrected it to "ND" (not determined) and added the footnote in Table 2.

  • Selectivity indices (SI) for anticancer active compounds should be calculated in each cell line.

Response 8: Thanks for the reviewer's suggestion. According to the reviewer's comments, we have added the selectivity index (SI) to analyze the sensitivities of each compound against cancer cell lines (these results are listed in the new Table 3.). The contents regarding SI are added in the revised manuscript.

  • Fig. 7: The bioavailability radar graphs as well as the boiled egg prediction model are not of good enough quality to read.

Response 9: Thanks for the reviewer's comment. We have rearranged the bioavailability radar graphs and the boiled egg prediction model in Figure 7 and saved them in JPG format with high-resolution pixels.

  • Section 3.3.: The concentration range in which the compounds were tested should be given

Response 10: According to the reviewers' comments, we have added the sentence regarding the testing concentrations range as "The tested compounds were initially prepared in the stock solution with a concentration of 100 μM and diluted to evaluate until found the IC50 values. The IC50 values of more than 50 μM are regarded as having no anti-proliferation effect." into 3.3. MTT Assay section.

  • References should be adapted to the journal's requirements.

Response 11: Thanks for the reviewer's comment. We checked the reference formats and edited them with EndNote X7.

  • Text formatting should be carefully checked.

Response 12: We thank the reviewer's reminder. We have checked the text formatting according to the submission guideline and utilized the MDPI Microsoft Word template.

  • The language should be modified carefully.

Response 13: Thanks for the reviewer's comment. We have rewritten the revised manuscript carefully and checked the grammar on the English writing website.

Sincerely,

First author

Yi-Fong Chen, Ph.D.

Full Current Mailing Address:

Graduate Institute of Cancer Biology and Drug Discovery

College of Medical Science and Technology

Taipei Medical University

E-mail: slugcousin7682@gmail.com

      slugcousin7682@tmu.edu.tw